# Role of Microaneurysms in the Pathogenesis and Therapy of Diabetic Macular Edema: A Descriptive Review

**DOI:** 10.3390/medicina59030435

**Published:** 2023-02-22

**Authors:** Yoshihiro Takamura, Yutaka Yamada, Masaru Inatani

**Affiliations:** Department of Ophthalmology, Faculty of Medical Sciences, University of Fukui, Yoshida 910-1193, Japan

**Keywords:** diabetic retinopathy, diabetic macular edema, microaneurysms, retina, photocoagulation, VEGF, anti-vascular endothelial growth factor

## Abstract

*Background and Objectives:* This study aims to elucidate the role of microaneurysms (MAs) in the pathogenesis and treatment of diabetic retinopathy (DR) and diabetic macular edema (DME), the major causes of acquired visual impairment. *Materials and Methods:* We synthesized the relevance of findings on the clinical characteristics, pathogenesis, and etiology of MAs in DR and DME and their role in anti-vascular endothelial growth factor (VEGF) therapy. *Results:* MAs, a characteristic feature in DR and DME, can be detected by fluorescein angiography, optical coherence tomography (OCT) and OCT angiography. These instrumental analyses demonstrated a geographic and functional association between MA and ischemic areas. MA turnover, the production and loss of MA, reflects the activity of DME and DR. Several cytokines are involved in the pathogenesis of MAs, which is characterized by pericyte loss and endothelial cell proliferation in a VEGF-dependent or -independent manner. Ischemia and MAs localized in the deep retinal layers are characteristic of refractory DME cases. Even in the current anti-VEGF era, laser photocoagulation targeting MAs in the focal residual edema is still an effective therapeutic tool, but it is necessary to be creative in accurately identifying the location of MAs and performing highly precise and minimally invasive coagulation. *Conclusions:* MAs play a distinctive and important role in the pathogenesis of the onset, progression of DR and DME, and response to anti-VEGF treatment. Further research on MA is significant not only for understanding the pathogenesis of DME but also for improving the effectiveness of treatment.

## 1. Introduction

Diabetic retinopathy (DR) and diabetic macular edema (DME) are the major causes of acquired visual impairment with the background of the currently growing number of the patients with diabetes mellitus [1]. Microaneurysms (MAs), formed by proliferating endothelial cells (ECs) and pericytes loss due to chronic hyperglycemia, is a typical feature in DR and DME and can be a marker indicating their activity [2]. The treatment of DR and DME has traditionally involved photocoagulation for retinal ischemic areas and Mas; however, it has currently shifted to anti-VEGF therapy [1]. To date, ranibizumab, aflibercept, brolucizumab, and faricimab have been approved for the treatment of DME in Japan. Frequently repeated injections of intravitreal anti-VEGF agents have the promising effects in improving visual acuity and retinal thickness; nevertheless, it is reported that there are still 40% of cases that are refractory with poor response to this treatment [3]. Many studies have reported that MA plays an important role in the pathogenesis of DR and DME and their response to anti-VEGF therapy. In this paper, we summarize the important findings within the literature on MAs concerning the pathogenesis and clinical management of DR and DME.

## 2. Clinical Feature of Microaneurysms

### 2.1. Diabetic Retinopathy (DR)

DR results from changes in retinal microvascular structures and blood flow due to persistent hyperglycemia [4]. MAs are usually the earliest manifestations of DR and appear as tiny red dots scattered throughout the retina as well as the hallmark of clinical diagnosis of DR. If the MAs that leak were not present in the macular area, MAs do not have any manifestations and do not affect the vision of patients; however, early recognition of MAs can lead to early detection and treatment of DR that will reduce the likelihood of vision loss. MAs cannot be distinguished from tiny dot hemorrhages, and they can be occasionally undetectable ophthalmoscopically. For this reason, fluorescein angiography (FA) has been considered the gold standard for the detection of MAs, visualized as hyperfluorescent dots on early phase, while the dot hemorrhages are visualized as hypofluorescent dots (Figure 1) [5]. FA is sensitive to detect even very small MAs, and the detection rate varies depending on the phase of angiography, with 33% reported only in the early phase, 31% reported in the late phase, and the rest reported MAs detected continuously from early to late phase [6]. However, FA is not suitable for frequent screening for DR because of the potential side effects of fluorescence dye, including as nausea, vomiting, and shock. Optical coherence tomography (OCT) is a complementary modality that can visualize MAs noninvasively. OCT B scan depicts MAs as a well-defined intraretinal hyperreflective lesion with a circular or oval border [7]. MAs can also be observed by OCT angiography (OCTA), but its detection sensitivity is inferior to that of FA; OCTA only detected 41.0 ± 16.1% of MAs in FA images [8]. The reason for this may be that OCTA is limited by the principle of slowest detectable flow. Multiple image averaging may help increase the MA detection capability of OCTA especially for focal bulge-type MAs [9]. Moreover, OCTA cannot depict the leakage from blood vessels and MAs and thus cannot evaluate the degree of the vascular permeability.

### 2.2. DME

Diabetic macular edema (DME), which can be classified as focal or diffuse based on FA findings, is the leading cause of visual impairment in patients with DR [10]. DME can occur at any stage of DR, and macular edema with the leakage from MAs can cause vision loss. Prolonged hyperglycemia leads to chronic damage of the retinal microvasculature and hypoxia, resulting in increased intraocular concentrations of vascular endothelial growth factor (VEGF) and elevated vascular permeability [11]. MAs and compromised vessels due to the disruption of the blood–retinal barrier (BRB) are the main cause of focal and diffuse edema, respectively [1]. Typical of focal DME, edema is formed by leakage from MAs, eventually forming hard exudates consisting of lipoprotein residues of serous leakage from damaged vessels and MAs. The accumulation of pin-point leakage from MAs is detected in the early phase, while diffuse leakage starts from the early phase and persists until the late phase. Automatic grading is helpful for the quick diagnosis of DME according to the shortest distance from the hard exudates to the fovea [12,13]. Diffuse DME is also characterized by MAs and capillary dropouts (CDOs), which are hypofluorescent regions representing focal ischemia. When clinicians are asked to classify DME as focal or diffuse, a wide variation is noted among the studies, indicating that it is clinically difficult to distinguish between focal and diffuse DME [14]. As new pathogenetic classification defined by Parodi et al., four subtypes of DME can be identified: vasogenic (DME with vascular dilation), nonvasogenic (DME without vascular dilation), tractional and mixed DME [15]. Relative indications include laser irradiation, especially for the vasogenic subform of DME, which is clinically characterized by the presence of focally localized MA and leaky capillaries [16].

### 2.3. Distribution Pattern of MAs in Diffuse DME

Using ultra-widefield FA, a study showed that the larger the area of non-perfused retina and the greater the severity of DR, the more likely it is to be diffuse DME; conversely, the smaller the level of ischemia, the greater the possibility of focal DME [17]. This finding suggests that the type of DME progresses from focal to diffuse as retinal ischemia worsens. MAs are usually seen in the border areas of CDOs in DR [18]; this association results in a characteristic distribution of MAs in diffuse DME (Figure 2) [19]. There were more MAs in the periphery than in the central area of the edema. Although CDOs in the periphery of the edema are small, they have a large circumference and have fine irregularities and fragmentation. Furthermore, since there are more CDOs in the periphery of the edema, there are also more MAs adjacent to them as compared to CDOs in the center of the edema [19]. Active leakage from several MAs in the edema periphery would contribute to the expansion of edematous areas. In fact, severe ischemia leads to large size edema that extends beyond the macular area [20].

OCTA demonstrated that approximately 80% of MAs are distributed in the deep capillary plexuses (DCP) of the retina [21]. Horii et al. also reported that 80.3% of the MAs were located in the inner nuclear layer, which contains the DCP [22]. Ishibazawa et al. used OCTA to examine diabetic eyes and showed that MAs were mainly located in the DCP [23]. Anatomically, the majority of capillaries lie in the DCP and are surrounded only by pericytes. Hence, it is reasonable that most of the MAs are in the DCP. The increased number of MAs in the DCP is associated with macular volume, and thus, the MAs in the DCP contribute to the pathogenesis of DME, especially the cystoid type [21]. Direct photocoagulation is an effective treatment for MAs, but coagulation with excessive power that reaches the photoreceptors and pigment epithelium should be avoided. It is important to accurately determine the location of MAs in the retina to prevent tissue damage during laser therapy.

## 3. Pathology of MAs in DR and DME

### 3.1. VEGF May Potentially Induce the Development of MAs

MA formation may be induced by intravitreal injection of VEGF as demonstrated by a study using monkey eyes [24]. This finding suggests that the formation of MAs with leakage may be an indicator of VEGF-mediated pathology. The development of MAs may also be enhanced by CDOs, which is a possible source of VEGF, near the MAs. We showed that approximately 80% of MAs were adjacent to CDOs [19]. Dilated shunt vessels with MAs are adjacent to non-perfused acellular capillaries in the retina of patients with DR. It is possible that vessel dilation results in imbalances in fluid flow and viscosity, resulting in the further formation of dilated shunt vessels and MAs neighboring non-perfused capillaries [18]. Contrary to the induction of MA by VEGF, anti-VEGF treatment of DME reduces the number of MAs. Sugimoto et al. reported that the reduction rate of MAs after three times injections of aflibercept was 50.4 ± 21.2% [25]. Therefore, VEGF is thought to be involved in the pathogenesis of both the production and loss of MAs.

In the mouse retina, the loss of pericytes causes endothelial inflammation and perivascular macrophage infiltration [26,27]. Macrophage-derived VEGF activates the VEGF-receptor 2 in ECs. Furthermore, in Mas, ECs without pericytes cause an elevation of angiopoietin-2 (Ang2) levels [26,27]. In normal vessels, vascular stability and homeostasis are maintained by Ang1-tyrosine-protein kinase receptor (Tie2) signaling [28]. However, in pathological vessels, Ang2 inhibits Ang1 and Tie2 receptor binding and destabilizes vessels, weakening EC junctions and leading to vascular leakage. Pericyte loss further destabilizes vessels and induces MA formation and neovascularization. 

ANG/Tie2 signaling is functional in pericytes and plays an important role in DR progression. In pericytes undergoing apoptosis induced by hyperglycemia, Ang1 promotes cell survival, whereas Ang2 promotes apoptosis [29]. Under normal glycemic conditions, Ang1 or Ang2 does not exert any influence on apoptotic cells. On the other hand, the overexpression of Ang2 in nondiabetic retinas mimicked diabetic pericyte migration in wild-type mice, while a hyperglycemia-induced increase in pericyte migration was not observed in Ang2-deficient mice [30]. Moreover, Park et al. demonstrated that Ang2 induces pericyte apoptosis via the p53 pathway under hyperglycemic conditions, whereas Ang2 alone does not [31]. Integrin is also involved in Ang2-induced pericyte apoptosis under hyperglycemic conditions, as it acts as an Ang2 receptor. These findings suggest that blocking Ang2 signaling could be a potential therapeutic target to prevent pericyte loss in early DR. The concentration of Ang2 in the vitreous of the patients with DR has been reported to be higher compared to control (macular hole) [32]. Recently, faricimab, a bispecific antibody targeting Ang2 and VEGF, was approved for the treatment of DME [33]. YOSEMITE and RHINE trials demonstrated that anatomical and visual improvement with faricimab were achieved and has a potential to extend the dosing interval up to 16 weeks for the patients with DME [34]. Faricimab may decrease the production and total number of MAs during the course of DME treatment. A sample case is shown in Figure 3. After 3 times monthly injection of faricimab, the total number of MAs decreased from 143 to 52. In the treated eye, the ratios of new developed MAs in MA turnover (the total number of appeared and disappeared MAs) were only 7.5% (8/107), while there were 57.6% (38/66) in the untreated other eye. Further studies are needed to determine how faricimab affects MA dynamics. 

### 3.2. Mechanisms of Pericyte Dropout 

The theory that VEGF produced by ischemia induces MA formation does not explain why MAs are the first morphological abnormality in DR as there is also a VEGF-independent mechanism for MA formation. MAs are accompanied by endothelial hyperplasia resulting from aberrant proliferation, basement membrane thickening, and a decreased number of pericytes [2,35]. Pericyte loss occurs in both diabetic and galactose-fed dogs and is characterized by changes in retinal vessels, such as MAs, hemorrhage, and the formation of non-perfused areas, similar to those seen in human DR [36,37,38]. Experimental evidence suggests that these changes can be prevented by aldose reductase inhibitors (ARI) [36,39,40]. Excess sugar alcohol (polyol) accumulation, catalyzed by the enzyme aldose reductase (AR) during hyperglycemic conditions, has been implicated in the pathogenesis of DR. Immunohistochemical studies have demonstrated that there is abundant AR expression in pericytes isolated by trypsin digestion from human and dog retinas [41]. Apoptosis was induced in pericytes cultured in a high-glucose medium and was prevented by treatment with ARI [40,42], suggesting that the polyol pathway via AR plays an important role in the pathogenesis of pericyte loss.

Apoptosis was not observed in galactose-exposed retinal ECs that have low AR content and activity. On the other hand, AR-overexpressing ECs showed decreased cell viability and polyol accumulation, similar to that in pericytes. This suggests that the physiological difference in response to hyperglycemia is attributable to the level of AR expression and is not a cell-specific feature of pericytes and ECs. A study employing a co-culture system of pericytes and ECs exposed to a high-glucose medium demonstrated that there was an increased proliferation of ECs as the number of pericytes decreased. Biochemical assays disclosed that the levels of active transforming growth factor-beta (TGF-β) in media were linked to EC growth. Supplying active TGF-β to a co-culture medium containing high-glucose restored the inhibitory activity against EC growth. 

Platelet-derived growth factor (PDGF)-B-deficient mouse embryos showed a lack of microvascular pericytes, resulting in numerous MAs [43]. However, PDGF-B/PDGF receptor β signaling did not cause disruption of the BRB in adult mice [26]. In the pathogenesis of pericyte loss and EC proliferation, various factors are intricately related to each other.

## 4. Clinical Role of MAs in the Management of DR and DME

### 4.1. MA Turnover Is a Biomarker for Disease Activity and Treatment

MAs do not remain stable in the retina in DR and DME for long periods of time. The appearance and disappearance of MAs, defined as MA turnover, represent a dynamic process and reflects disease activity, and it can be a predictor of DR and DME progression. A 5-year prospective longitudinal study demonstrated that MA turnover and MA formation rates are related to the development of vision-threatening complications, such as DME and proliferative DR, and the worsening of DR [44]. Additionally, RetmarkerDR analysis showed that mild non-PDR eyes with lower MA turnover are less likely to develop DME in 2 years [45]. Haritoglou et al. reported that an increased MA formation rate of 2 or more was associated with the development of DME [46]. MA turnover is also influenced by anti-VEGF treatment. The injections of anti-VEGF drug ranibizumab reduce macular thickness and enhance MA turnover in DME; the number of MAs that disappeared was more than those that developed, and the absolute number of MAs also decreased after treatment [47]. Therefore, MA turnover can be a biomarker for treatment response as well as disease activity. Frequent examinations to detect MAs before and after injection are required to evaluate MA turnover. In this respect, OCTA may be suitable for monitoring MA turnover, since this tool has no complications due to intravenous injection of fluorescein dye, although it is less sensitive than FA in detecting MA.

### 4.2. MAs Is Associated with Resistance to Anti-VEGF Therapy

In DME treatment, anti-VEGF therapy is effective for reducing the retinal thickness and decreasing the size of edematous areas; however, residual focal edema frequently remains, as seen in 65.8% of cases after the first injection [48]. An analysis using a 3D mode OCT map wherein an edematous area was divided into 100 sections showed that the reduction in retinal thickness after anti-VEGF therapy varied in regions of the DME [49]. A 10–20% reduction in retinal thickness accounted for approximately 40% of the total edematous areas, whereas only 6.4% of the edematous areas showed a reduction in retinal thickness of 30% or more. Areas with a reduction in retinal thickness of less than 5% were indicative of refractoriness to anti-VEGF therapy, and they accounted for approximately 10% of the edematous areas. These results suggest that the edema-improving effect of anti-VEGF therapy varies by site and that some sites are less responsive than others. Notably, the MA density was higher in refractory areas, which was also where the reduction in retinal thickness after anti-VEGF therapy was minimal [49]. Additionally, the areas with residual edema after anti-VEGF treatment had a higher density of MAs than areas with improvement in edema [48].

DME due to MAs is classified as focal DME and is distinguished from diffuse DME due to extensive permeability enhancement [1]. The therapeutic efficacy of anti-VEGF drugs tends to be lower in focal DME than in diffuse DME [50]. Hirano et al. showed that the number of ranibizumab injections required to obtain stable improvement of edema is significantly higher when MAs are present in the areas adjacent to the foveal avascular zone (FAZ) [39,51]. Lee et al. also reported that small vascular flow density in the DCP, large FAZ area, and large number of adjacent MAs are characteristic of cases with poor response to anti-VEGF therapy [52]. These findings support MAs as a contributory factor to resistance to anti-VEGF therapy.

### 4.3. Direct Photocoagulation Aiming MAs

After anti-VEGF injection into the areas involved in DME, the appearance of the fovea usually returns to almost normal; however, focal edema often persists in the paracentral area. If injections are discontinued because edema has improved in the central area and results in improved visual acuity, the residual perifoveal edema may expand and affect the central areas, as shown by the sample case in Figure 4. Hence, MAs within the residual edema should be targets of additional treatment.

If high MA density is a risk factor for residual edema after anti-VEGF therapy, what treatment strategy should ophthalmologists use? We reported that repeated additional injections of anti-VEGF agents for residual edema with high-density MA resulted in edema resolution in approximately 90% of cases [48]. However, significantly more injections were required than in the case group that had no residual edema after a single injection [48]. Although more injections are required, they can improve edema even in cases refractory to anti-VEGF therapy. 

Guidelines for the management of DME by the European Society of Retina Specialists (EURETINA) stated that laser photocoagulation is no longer recommended for the treatment of DME, and anti-VEGF therapy has emerged as first-line therapy [16]. Nevertheless, direct laser photocoagulation that targets MAs may also be an alternative treatment modality for the residual focal DME after anti-VEGF therapy with cluster of MAs which locates outside the fovea. Although it is difficult to precisely identify MAs, compositing FA images depicting MAs with OCT maps and fundus photographs improves outcomes of focal laser therapy for focal DME [53]. Alternatively, Mori et al. reported that DME recurrence is more common in patients with significantly higher numbers of MAs in the late phase of indocyanine green angiography (IA) [54]. MAs depicted by IA are often localized within the edema and have the advantage of being easy targets for laser because of their low leakage compared to FA [5]. Recently, navigational laser systems have been developed to automatically apply precise laser therapy to MA [5,55]. The navigated laser photocoagulation is performed based on planned treatment locations according to the real-time fundus image. The navigated laser has an eye-tracking laser delivery system and allows more accurate focal laser photocoagulation than conventional focal laser therapy for DME. Further clinical studies are needed to determine the efficacy of these laser treatments for residual macular edema.

Excessive conventional laser therapy has several complications such as night vision, the impairment of contrast and visual-field sensitivity, choroidal neovascularization, and the enlargement of laser scars [16]. The subthreshold micropulse laser (SMPL) is a relatively new retinal laser technology that has proven to be safer for retinal tissue than conventional continuous wavelength lasers. SMPL hardly induces the formation of retinal scars and retinal damages, and several studies have shown that this SMPL is effective treatment for DME, in terms of the improvement of visual function and the retinal thickness. In addition, Vujosevic et al. showed that SMPL decreased the number of MAs at 3 months in the DCP, at 6 and 12 months in both deep and superficial capillary plexuses (SCP) [56]. They also showed that SMPL induces more pronounced changes in the DCP than in the SCP in DME [57]. DCP is a preferred site for MAs, and thus, SMPL may allow laser therapy specifically related to the pathogenesis of MAs in the treatment for DME. 

## 5. Conclusions

MAs, the earliest pathological changes observed in DR, are accompanied by pericyte loss and EC proliferation. Relating to retinal ischemia, several cytokines such as VEGF, ANG-2 and TGF-beta are associated with the synthesis of MAs and pathology including leakage. In diffuse DME, MAs frequently develop in the periphery of edema, and leakage from MAs may contribute to edema expansion. The effectiveness of anti-VEGF agents is relatively less for the MAs. High-dense MAs were observed in areas where there was residual focal edema and where retinal thickness was minimally reduced after anti-VEGF treatment. Although the repeated injections of anti-VEGF agent are gold standard, direct photocoagulation that targets MAs in residual focal edema after anti-VEGF therapy is also effective, and several efforts have been attempted to improve therapeutic outcomes. MAs play a distinctive and important role in the pathogenesis of the onset, progression, and treatment response in DR and DME. Further basic and clinical research on MAs is needed to discuss the future prospects and unmet needs for clinical management of DR and DME.

## Figures and Tables

**Figure 1 medicina-59-00435-f001:**
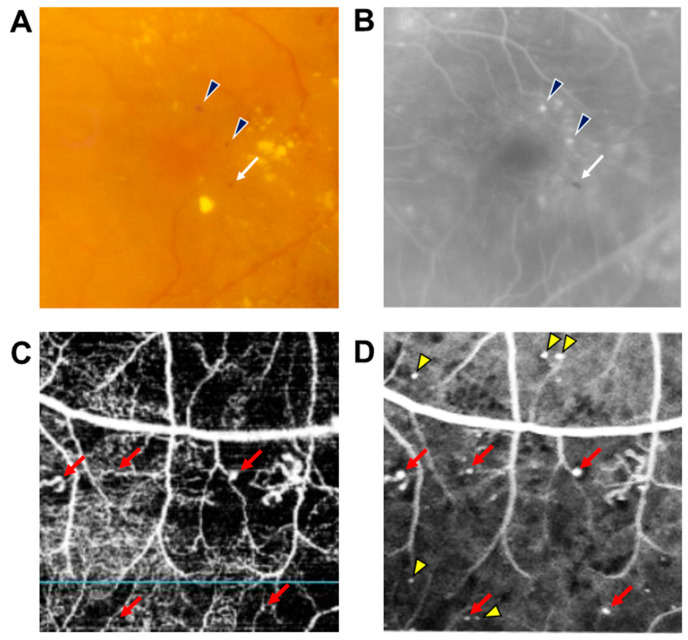
Microaneurysms detected by fluorescein angiography and optical coherence tomography angiography (OCTA). (**A**) Both microaneurysms (MAs) and dot hemorrhages are shown as red dots on fundus photographs. (**B**) MAs are depicted as hyperfluorescent (black arrowheads), and dot are depicted as hemorrhages as hypofluorescent (white arrows) on fluorescein angiography (FA). In the detection of MAs by OCTA (**C**), some MAs depicted by FA (**D**) are detected (red arrows), while others are not (yellow arrowheads).

**Figure 2 medicina-59-00435-f002:**
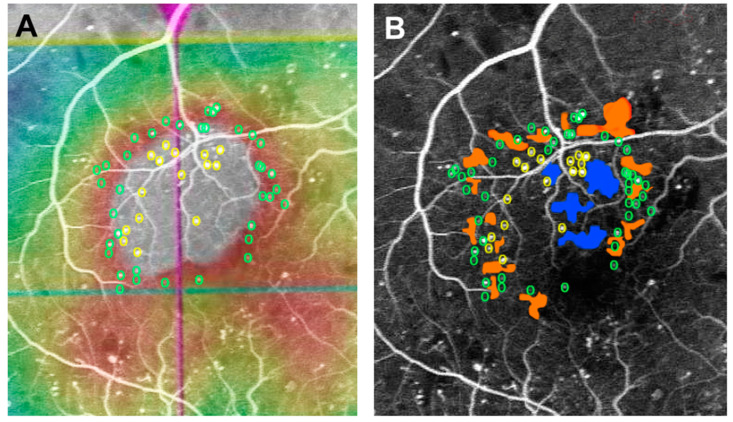
Marking of microaneurysms and capillary dropouts in a merged image. (**A**) Distribution of microaneurysms (MAs) in the center (yellow circle) and periphery (green circle) of diabetic macular edema (DME). (**B**) Distribution of the capillary dropouts (CDOs) in the center (blue area) and periphery (orange area) of DME.

**Figure 3 medicina-59-00435-f003:**
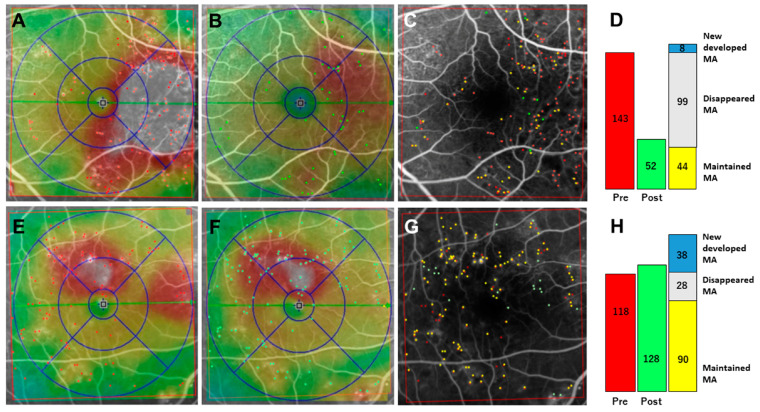
A sample case showing microaneurysm turnover after faricimab treatment. Fluorescein angiography (FA) image was merged with OCT map in the eye treated with faricimab (**A**–**C**). MAs before (**A**) and after (**B**) faricimab treatment and were filled in red and green, respectively, and these figures were merged (**C**). Images taken at same time point in another untreated eye were shown in (**E**–**G**). Color bar indicates the number of MAs before (red), after (green), new developed (blue), disappeared (gray), and maintained (yellow) in the eyes treated (**D**) and untreated (**H**) with faricimab.

**Figure 4 medicina-59-00435-f004:**
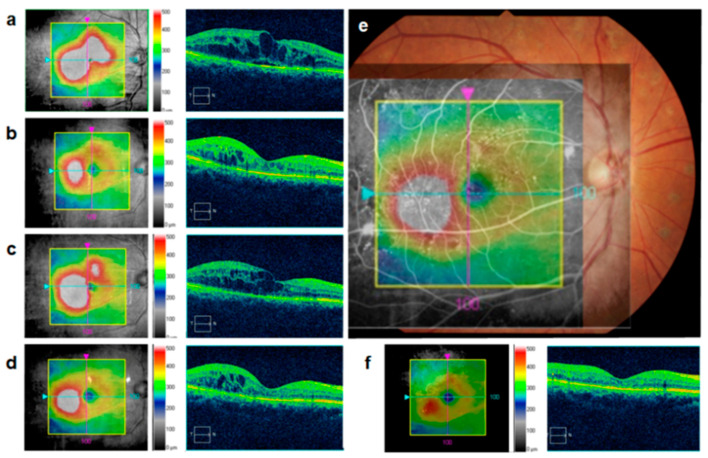
Microaneurysms in areas of residual edema after anti-VEGF treatment. (**a**) Optical coherence tomography (OCT) map and cross-sectional images show a representative case of diabetic macular edema (DME) and its improvement (**b**,**d**) and recurrence (**c**) after anti-VEGF treatment. (**e**) Merged images show microaneurysms (MAs) in areas of residual focal edema. (**f**) After direct photocoagulation aiming MAs, focal edema improved, and recurrence was not observed.

## Data Availability

The datasets generated during and/or analyzed during the current study are available from the corresponding author on reasonable request.

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
