# Peer review of "Role of Microaneurysms in the Pathogenesis and Therapy of Diabetic Macular Edema: A Descriptive Review"

_medicina, 2023, doi:10.3390/medicina59030435_

Round 1

Reviewer 1 Report

The review is well organised and of clear interest.

Congratulations to the Author for the overall quality and interest of the review.

The relationship of new MA formation and areas of capillary dropout is clearly a major and relevant question and is addressed, although briefly.

Author Response

Reviewer 1

Comments and Suggestions for Authors

The review is well organized and of clear interest.

Congratulations to the Author for the overall quality and interest of the review.

The relationship of new MA formation and areas of capillary dropout is clearly a major and relevant question and is addressed, although briefly.

We deeply appreciate the reviewer’s high evaluation.

Reviewer 2 Report

This paper presents that microaneurysms (MAs), the earliest pathological changes observed in diabetic retinopathy (DR), are accompanied by pericyte loss and endothelial cell (EC) proliferation. In DME,  MAs frequently develop in the periphery of edema, and leakage from MAs may contribute to edema expansion. Intravitreal injection of anti-VEGF agents is effective for DME; however, it is less effective for MAs. When MAs are seen in the perifoveal area, more injections are required to achieve significant improvement of edema. Several MAs were observed in areas where there was residual focal edema and where retinal thickness was minimally reduced after anti-VEGF treatment. MA turnover is a biomarker of disease activity and treatment. Direct photocoagulation that targets MAs in residual focal edema after anti-VEGF therapy is effective, and several efforts have been attempted to improve therapeutic outcomes. Further research on MA is significant not only for understanding the pathogenesis of DME but also for improving the effectiveness of treatment.

Some papers may be related to this topic:

(1) Automatic grading of Diabetic macular edema based on end-to-end network Fovea localization by blood vessel vector in abnormal fundus images;

(2) Fovea localization by blood vessel vector in abnormal fundus images;

Author Response

Comments and Suggestions for Authors

This paper presents that microaneurysms (MAs), the earliest pathological changes observed in diabetic retinopathy (DR), are accompanied by pericyte loss and endothelial cell (EC) proliferation. In DME, MAs frequently develop in the periphery of edema, and leakage from MAs may contribute to edema expansion. Intravitreal injection of anti-VEGF agents is effective for DME; however, it is less effective for MAs. When MAs are seen in the perifoveal area, more injections are required to achieve significant improvement of edema. Several MAs were observed in areas where there was residual focal edema and where retinal thickness was minimally reduced after anti-VEGF treatment. MA turnover is a biomarker of disease activity and treatment. Direct photocoagulation that targets MAs in residual focal edema after anti-VEGF therapy is effective, and several efforts have been attempted to improve therapeutic outcomes. Further research on MA is significant not only for understanding the pathogenesis of DME but also for improving the effectiveness of treatment.

The Reviewer's comments are very important and are quoted in the conclusion. “MAs, the earliest pathological changes observed in DR, are accompanied by pericyte loss and EC proliferation. Relating to retinal ischemia, several cytokines such as VEGF, ANG-2 and TGF-beta are associated with the synthesis of MAs and pathology including leakage. In diffuse DME, MAs frequently develop in the periphery of edema, and leakage from MAs may contribute to edema expansion. Effectiveness of anti-VEGF agents is relatively less for the MAs. High dense MAs were observed in areas where there was residual focal edema and where retinal thickness was minimally reduced after anti-VEGF treatment. Although the repeated injections of anti-VEGF agent are gold standard, direct photocoagulation that targets MAs in residual focal edema after anti-VEGF therapy is also effective, and several efforts have been attempted to improve therapeutic outcomes. MAs play a distinctive and important role in the pathogenesis of the onset, progression, and treatment response in DR and DME. Further basic and clinical research on MAs is needed to discuss the future prospects and unmet needs for clinical management of DR and DME.” (P.9, lines 17 to 30)

Some papers may be related to this topic:

(1) Automatic grading of Diabetic macular edema based on end-to-end network Fovea localization by blood vessel vector in abnormal fundus images;

(2) Fovea localization by blood vessel vector in abnormal fundus images;

We added the following sentence and 2 references in the revised version. “Automatic grading is helpful to quick diagnosis of DME according to the shortest distance from the hard exudates to the fovea.”  (P.3, lines 19 and 20)

Reviewer 3 Report

The manuscript entitled “Role of microaneurysms in the pathogenesis and therapy of diabetic macular edema” summarizes pivotal roles of microaneurysm in diabetic macular edema. This work provides useful information about how to treat diabetic macular edema and what the mechanism is for refractory diabetic macular edema to anti-VEGF therapy. However, I was having some difficulties in reading this article. Please discuss the issues below.

In my eyes, figure 1 does not seem to provide any useful information for understanding microaneurysm. The author mentioned about OCTA findings and cited the reference about focal bulge-type MAs. How about using both FA and OCTA findings of MAs? That should be helpful for readers. 

In figure 2, the abbreviation of microaneurysms should be MAs.

Page 3, the author stated the importance of determining the accurate location of MAs to avoid tissue damage by laser. Are there any tips to do so?

Page 5 line 7-9, I still do not understand why OCTA accompanied with FA is useful for detecting MA turnover. Please describe this reason and the definition of MA turnover.

Page 6, additional injection of anti-VEGF agents achieved edema resolution in 90% of cases. In other words, 10% cases were refractory to this therapy. Which should we do for this case, more injection or laser ablation?

Author Response

Reviewer 3

Comments and Suggestions for Authors

The manuscript entitled “Role of microaneurysms in the pathogenesis and therapy of diabetic macular edema” summarizes pivotal roles of microaneurysm in diabetic macular edema. This work provides useful information about how to treat diabetic macular edema and what the mechanism is for refractory diabetic macular edema to anti-VEGF therapy. However, I was having some difficulties in reading this article. Please discuss the issues below.

In my eyes, figure 1 does not seem to provide any useful information for understanding microaneurysm. The author mentioned about OCTA findings and cited the reference about focal bulge-type MAs. How about using both FA and OCTA findings of MAs? That should be helpful for readers. 

In accordance to reviewer’s suggestion, we added the following sentences (P. 2 lines 18 to 22)

“MA cannot be distinguished from tiny dot hemorrhages, and be occasionally undetectable ophthalmoscopically. For this reason, fluorescein angiography (FA) has been considered the gold standard for detection of MAs, visualized as hyperfluorescent dots on early phase, while the dot hemorrhages are visualized as hypofluorescent dots (Figure 1) [5].”

We also changed Figure 1 to show the images of FA and OCTA in the same eye. We changed the legend (underline) of Figure 1 as follows:

Figure 1. Microaneurysms detected by fluorescein angiography and optical coherence tomography angiography (OCTA). (A) Both microaneurysms (MAs) and dot hemorrhages are shown as red dots on fundus photographs. (B) MAs are depicted as hyperfluorescent (black arrowheads), and dot are depicted as hemorrhages as hypofluorescent (white arrows) on fluorescein angiography (FA). In the detection of MAs by OCTA (C), some MAs depicted by FA (D) are detected (red arrows), while others are not (yellow arrowheads).

In figure 2, the abbreviation of microaneurysms should be MAs.

 We changed from “Mas” to “MAs” in figure legend.

Page 3, the author stated the importance of determining the accurate location of MAs to avoid tissue damage by laser. Are there any tips to do so?

In response to the reviewers' questions, we wrote the following Discussion. (P. 8 lines 23 to P.9 line 2)

“Although it is difficult to precisely identify MAs, compositing FA images depicting MAs with OCT maps and fundus photographs improves outcomes of focal laser therapy for focal DME [53]. Alternatively, Mori et al. reported that DME recurrence is more common in patients with significantly higher numbers of MAs in the late phase of indocyanine green angiography (IA) [54]. MAs depicted by IA are often localized within the edema and have the advantage of being easy targets for laser because of their low leakage compared to FA [5]. Recently, navigational laser systems have been developed to automatically apply precise laser therapy to MA [5,55]. The navigated laser photocoagulation is performed based on planned treatment locations according to the real-time fundus image. The navigated laser has eye-tracking laser delivery system and allows more accurate for focal laser photocoagulation than conventional focal laser therapy for DME. Further clinical studies are needed to determine the efficacy of these treatments for residual macular edema.”

Page 5 line 7-9, I still do not understand why OCTA accompanied with FA is useful for detecting MA turnover. Please describe this reason and the definition of MA turnover.

We added the phrase “defined as MA turnover,” and wrote “MAs do not remain stable in the retina in DR and DME for long periods of time. The appearance and disappearance of MAs, defined as MA turnover, represent a dynamic process and reflects disease activity, and it can be a predictor of DR and DME progression.”  (P.7, lines 7 to 9)

We apologize the description was due to our careless mistake. "OCTA with intravenous injection of fluorescein dye" should be "OCTA without intravenous injection of fluorescein dye".

We changed sentence as follows (P.7, lines 20 to 23):

Frequent examinations to detect MAs before and after injection are required to evaluate MA turnover. In this respect, OCTA may be suitable for monitoring MA turnover, since this tool has no complications due to intravenous injection of fluorescein dye, though it is less sensitive than FA in detecting MA.

Page 6, additional injection of anti-VEGF agents achieved edema resolution in 90% of cases. In other words, 10% cases were refractory to this therapy. Which should we do for this case, more injection or laser ablation?

We added the following sentences (P.8, lines 18 to 23)Guidelines for the management of DME by the European Society of Retina Specialists (EURETINA) stated that laser photocoagulation is no longer recommended for the treatment of DME, and anti-VEGF therapy has emerged as first-line therapy [16]. Nevertheless, direct laser photocoagulation that targets MAs may also be an alternative treatment modality for the residual focal DME after anti-VEGF therapy with cluster of MAs which locates outside the fovea.”  Then we wrote regarding the efficacy of laser therapy using merged method, IA guided laser, and navigated laser.

Reviewer 4 Report

I think that authors should decide about the form of this paper. Now, it is like a chapter in the book on diabetic retinopathy, not a review. It is informative, congratulations on the sections on pathology of MAs, but the rest needs some more work.

Specific comments

1.     Abstract : I suggest to provide a more structured abstract. Author should indicate the purpose of the review, database analyzed, factors analyzed and summarize results of the search in a systemized way. Now it is just a simple description of randomly selected concepts. In the review author should base on evidence provided by the studies.

2.     MAs can affect vision if they leak. They might be a source of ME that leads to decrease in BCVA. In the early background DR it is possible that they do not affect vison, however this is not a rule. MAs are the hallmark of DR. Please be precise.

3.     ICG is not a tool for detection of MAs. It is used for choroidal diseases. FA is the gold standard for DR diagnosis.

4.     Please describe the presentation of MAs on FA and OCT more precisely.

5.     Focal and diffuse is  just a one classification. It is ok to use it basing on FA. I would recommend to describe also vasogenic and non-vasogenic type of DME (Parodi et al.). It refers to MAs too. Please refer to EURETINA recommendations (Schmidt-Erfurth U, Garcia-Arumi J, Bandello F, Berg K, Chakravarthy U, Gerendas BS, Jonas J, Larsen M, Tadayoni R, Loewenstein A. Guidelines for the Management of Diabetic Macular Edema by the European Society of Retina Specialists (EURETINA). Ophthalmologica. 2017;237(4):185-222. doi: 10.1159/000458539. Epub 2017 Apr 20. PMID: 28423385)

6.     “Using ultra-widefield FA, a study showed that the larger the area of non-perfused retina and the greater the severity of DR, the more likely it is to be diffuse DME; conversely, the smaller the level of ischemia, the greater the possibility of focal DME” – please provide references.

7.     Fig. 2 should be of better quality.

8.     OCTA – limitations of visualization of MAs with OCTA should be noted.

9.     Laser photocoagulation in the macular area is not a standard of care any more (look at EURETINA recommendations). It should be noted. Please provide info on subthreshold micropulse lasers.

10.  Please note potential benefits of using  faricimab basing on recent papers published in Lancet.

11.  Direct LPC of MAs. Again. It has to be clearly stated that it is not a standard of care any more, as retinal scars may lead to decrease of BCVA and scotomas. Special types of DME with cluster of MAs outside the fovea can be treated in special situations. Please refer to EURETINA recommendations. Please note subthreshold micropulse as effective in reducing DME and also number of MAs without leaving scars . (Vujosevic S, Gatti V, Muraca A, Brambilla M, Villani E, Nucci P, Rossetti L, De Cilla' S. OPTICAL COHERENCE TOMOGRAPHY ANGIOGRAPHY CHANGES AFTER SUBTHRESHOLD MICROPULSE YELLOW LASER IN DIABETIC MACULAR EDEMA. Retina. 2020 Feb;40(2):312-321. doi: 10.1097/IAE.0000000000002383. PMID: 31972802., Vujosevic S, Toma C, Villani E, Brambilla M, Torti E, Leporati F, Muraca A, Nucci P, De Cilla S. Subthreshold Micropulse Laser in Diabetic Macular Edema: 1-Year Improvement in OCT/OCT-Angiography Biomarkers. Transl Vis Sci Technol. 2020 Sep 30;9(10):31. doi: 10.1167/tvst.9.10.31. PMID: 33062394; PMCID: PMC7533727.)

Author Response

Comments and Suggestions for Authors

I think that authors should decide about the form of this paper. Now, it is like a chapter in the book on diabetic retinopathy, not a review. It is informative, congratulations on the sections on pathology of MAs, but the rest needs some more work.

 Based on reviewer’s suggestion, we added the introduction section (P. 1, lines 31 to P. 2, lines 9). We deleted section numbers.

Diabetic retinopathy (DR) and diabetic macular edema (DME) are the major causes of acquired visual impairment with the background of the currently growing number of the patients with Diabetes mellitus. Microaneurysms (MAs), formed by proliferating endothelial cells (ECs) and pericytes loss due to chronic hyperglycemia, is a typical feature in DR and DME and can be a marker indicating their activity. Treatment of DR and DME has traditionally involved photocoagulation for retinal ischemic areas and MAs, however has currently shifted to anti-VEGF therapy. To date, ranibizumab, aflibercept, brolucizumab, and aflibercept have been approved for the treatment of DME in Japan. Frequently repeated injections of intravitreal anti-VEGF agents have the promising effects in improving visual acuity and retinal thickness, nevertheless it is reported that there are still 40% of cases that are refractory with poor response to this treatment. Many studies have reported that MA plays an important role in the pathogenesis of DR and DME and their response to anti-VEGF therapy. In this paper, we summarize the important findings within the literature on MAs concerning the pathogenesis and clinical management of DR and DME.

Specific comments

  1. Abstract: I suggest to provide a more structured abstract. Author should indicate the purpose of the review, database analyzed, factors analyzed and summarize results of the search in a systemized way. Now it is just a simple description of randomly selected concepts. In the review author should base on evidence provided by the studies.

In accordance to reviewer’s suggestion, we have re-written it to the structured abstract in the revised version.

Background and Objectives: To elucidate the role of microaneurysms (MAs) in the pathogenesis and treatment of diabetic retinopathy (DR) and diabetic macular edema (DME), the major causes of acquired visual impairment. Materials and Methods: We synthesized the relevance of findings on the clinical characteristics, pathogenesis, and etiology of MAs in DR and DME and their role in anti-vascular endothelial growth factor (VEGF) therapy. Results. MAs, a characteristic feature in DR and DME, can be detected by fluorescein angiography, optical coherence tomography (OCT) and OCT angiography. These instrumental analyses demonstrated a geographic and functional association between MA and ischemic areas. MA turnover, the production and loss of MA, reflects the activity of DME and DR. Several cytokines are involved in the pathogenesis of MAs, characterized by pericyte loss and endothelial cell proliferation, in a VEGF-dependent or -independent manner. Ischemia and MAs localized in the deep retinal layers are characteristic of refractory DME cases. Even in the current anti-VEGF era, laser photocoagulation targeting MAs in the focal residual edema is still an effective therapeutic tool, but it is necessary to be creative in accurately identifying the location of MAs and performing highly precise and minimally invasive coagulation. Conclusion. MAs play a distinctive and important role in the pathogenesis of the onset, progression of DR and DME, and response to anti-VEGF treatment. Further research on MA is significant not only for understanding the pathogenesis of DME but also for improving the effectiveness of treatment.

  1. MAs can affect vision if they leak. They might be a source of ME that leads to decrease in BCVA. In the early background DR it is possible that they do not affect vison, however this is not a rule. MAs are the hallmark of DR. Please be precise.

In accordance to reviewer’s comment, we added the following phrase, “as well as the hallmark of clinical diagnosis of DR. If the MAs that leak were not present in the macular area,” in the first paragraph (P.2, lines 15 and 16) of the revised version. Also, we added the following sentence “DME can occur at any stage of DR, and macular edema with the leakage from MAs can cause vision loss.”  (P. 3, lines 9 and 10).

  1. ICG is not a tool for detection of MAs. It is used for choroidal diseases. FA is the gold standard for DR diagnosis.

Based on reviewer’s comment, we deleted the phrase “or indocyanine green angiography (ICG)” and added “(FA), gold standard for diagnosis of DR” in the first paragraph.

  1. Please describe the presentation of MAs on FA and OCT more precisely.

Based on reviewer’s comment, we changed from “Although MAs may occasionally be undetected ophthalmoscopically, they may be detected by fluorescein angiography (FA) or indocyanine green angiography (ICG)” to “MAs cannot be distinguished from tiny dot hemorrhages, and be occasionally undetectable ophthalmoscopically. For this reason, fluorescein angiography (FA) has been considered the gold standard for detection of MAs, visualized as hyperfluorescent dots on early phases, while the dot hemorrhages are visualized as hypofluorescent dots (Figure 1). FA is sensitive to detect even very small MAs, and the detection rate varies depending on the phase of angiography, with 33% reported only in the early phase, 31% in the late phase, and the rest reported MAs detected continuously from early to late phase. However, FA is not suitable for frequent screening for DR because of the potential side effects of fluorescence dye, including as nausea, vomiting, and shock. Optical coherence tomography (OCT) is a complementary modality that can visualize MAs noninvasively. OCT B scan depicts MA as a well-defined intraretinal hyperreflective lesion with a circular or oval border.” (P.2, lines 18 to 29)

  1. Focal and diffuse is  just a one classification. It is ok to use it basing on FA. I would recommend to describe also vasogenic and non-vasogenic type of DME (Parodi et al.). It refers to MAs too. Please refer to EURETINA recommendations (Schmidt-Erfurth U, Garcia-Arumi J, Bandello F, Berg K, Chakravarthy U, Gerendas BS, Jonas J, Larsen M, Tadayoni R, Loewenstein A. Guidelines for the Management of Diabetic Macular Edema by the European Society of Retina Specialists (EURETINA). Ophthalmologica. 2017;237(4):185-222. doi: 10.1159/000458539. Epub 2017 Apr 20. PMID: 28423385)

In accordance to reviewer’s suggestion, we added the following sentences. “As new pathogenetic classification defined by Parodi et al., 4 subtypes of DME cab be identified: vasogenic (DME with vascular dilation), nonvasogenic (DME without vascular dilation), tractional and mixed DME. Relative indications include laser irradiation, especially for the vasogenic subform of DME, which is clinically characterized by the presence of focally localized MA and leaky capillaries.” (P.3, lines 24 to 29)

We added the following 2 references in the revised manuscript.

Parodi Battaglia M, Iacono P, Cascavilla M, Zucchiatti I, Bandello F. A Pathogenetic Classification of Diabetic Macular Edema. Ophthalmic Res 2018;60:23–8. https://doi.org/10.1159/000484350.

Schmidt-Erfurth U, Garcia-Arumi J, Bandello F, Berg K, Chakravarthy U, Gerendas BS, et al. Guidelines for the Management of Diabetic Macular Edema by the European Society of Retina Specialists (EURETINA). Ophthalmologica 2017;237:185–222. https://doi.org/10.1159/000458539.

  1. “Using ultra-widefield FA, a study showed that the larger the area of non-perfused retina and the greater the severity of DR, the more likely it is to be diffuse DME; conversely, the smaller the level of ischemia, the greater the possibility of focal DME” – please provide references.

We added the following reference in the revised version. “Patel RD, Messner L V., Teitelbaum B, Michel KA, Hariprasad SM. Characterization of ischemic index using ultra-widefield fluorescein angiography in patients with focal and diffuse recalcitrant diabetic macular edema. Am J Ophthalmol 2013;155:1038-1044.e2.”

  1. Fig. 2 should be of better quality.

Based on reviewer’s comment, we modified the Figure 2 in the revised manuscript.

  1. OCTA – limitations of visualization of MAs with OCTA should be noted.

We added the following sentence “The reason for this may be that OCTA is limited by the principle of slowest detectable flow.” (P.2, lines 32 and 33) Moreover, we changed the Figure 1 to show lower sensitivity of OCTA to detect MAs. In addition to the lower detection rate for MAs in the OCTA analysis, we added the following sentence as OCTA limitation. “Moreover, OCTA cannot depict the leakage from blood vessels and MAs, and thus cannot evaluate the degree of the vascular permeability.” (P.2, lines 35 and 36)

  1. Laser photocoagulation in the macular area is not a standard of care anymore (look at EURETINA recommendations). It should be noted. Please provide info on subthreshold micropulse lasers. 11.  Direct LPC of MAs. Again. It has to be clearly stated that it is not a standard of care anymore, as retinal scars may lead to decrease of BCVA and scotomas. Special types of DME with cluster of MAs outside the fovea can be treated in special situations. Please refer to EURETINA recommendations.

We agree the reviewer’s valuable comment, so we quoted and wrote as follows: “Guidelines for the management of DME by the European Society of Retina Specialists (EURETINA) stated that laser photocoagulation is no longer recommended for the treatment of DME, and anti-VEGF therapy has emerged as first-line therapy. Nevertheless, direct laser photocoagulation that targets MAs may also be an alternative treatment modality for the residual focal DME after anti-VEGF therapy with cluster of MAs which locates outside the fovea.” (P.8, lines 18 to 23)

Please note subthreshold micropulse as effective in reducing DME and also number of MAs without leaving scars . (Vujosevic S, Gatti V, Muraca A, Brambilla M, Villani E, Nucci P, Rossetti L, De Cilla' S. OPTICAL COHERENCE TOMOGRAPHY ANGIOGRAPHY CHANGES AFTER SUBTHRESHOLD MICROPULSE YELLOW LASER IN DIABETIC MACULAR EDEMA. Retina. 2020 Feb;40(2):312-321. doi: 10.1097/IAE.0000000000002383. PMID: 31972802., Vujosevic S, Toma C, Villani E, Brambilla M, Torti E, Leporati F, Muraca A, Nucci P, De Cilla S. Subthreshold Micropulse Laser in Diabetic Macular Edema: 1-Year Improvement in OCT/OCT-Angiography Biomarkers. Transl Vis Sci Technol. 2020 Sep 30;9(10):31. doi: 10.1167/tvst.9.10.31. PMID: 33062394; PMCID: PMC7533727.)

We added the following sentences as follows: (P. 9, lines 3 to 14)

“Excessive conventional laser therapy has several complications such as night vision, the impairment of contrast and visual-field sensitivity, choroidal neovascularization, and the enlargement of laser scars. The subthreshold micropulse laser (SMPL) is a relatively new retinal laser technology that has proven to be safer for retinal tissue than conventional continuous wavelength lasers. SMPL hardly induces the formation of retinal scars and retinal damages, and several studies shown that this SMPL is effective treatment for DME, in terms of the improvement of visual function and the retinal thickness. In addition, Vujosevic et al. showed that SMPL decreased the number of MAs at 3 months in the DCP, at 6 and 12 months in both deep and superficial capillary plexuses (SCP). They also showed that SMPL induces more pronounced changes in the DCP than in the SCP in DME. DCP is a preferred site for MAs, and thus SMPL may allow laser therapy specifically related to the pathogenesis of MAs in the treatment for DME.”

We added these reports in the references.       

  1. Please note potential benefits of using faricimab basing on recent papers published in Lancet.

In accordance to reviewer’s comment, we added the following sentences in the revised version.

The concentration of Ang-2 in vitreous of the patients with DR has been reported to be higher compared to control (macular hole). (P. 5, lines 27 and 28)

YOSEMITE and RHINE trials demonstrated that anatomical and visual improvement with faricimab were achieved and has a potential to extend the dosing interval up to 16 weeks for the patients with DME. (P. 5, lines 30 to 32)

Also, we added the following references.

Regula JT, Lundh von Leithner P, Foxton R, Barathi VA, Cheung CMG, Bo Tun SB, et al. Targeting key angiogenic pathways with a bispecific CrossMAb optimized for neovascular eye diseases. EMBO Mol Med 2016;8:1265–88.

Wykoff CC, Abreu F, Adamis AP, Basu K, Eichenbaum DA, Haskova Z, et al. Efficacy, durability, and safety of intravitreal faricimab with extended dosing up to every 16 weeks in patients with diabetic macular oedema (YOSEMITE and RHINE): two randomised, double-masked, phase 3 trials. Lancet (London, England) 2022;399:741–55. https://doi.org/10.1016/S0140-6736(22)00018-6.

Round 2

Reviewer 4 Report

It is a good revision. I have one comment - please name it a descriptive review (after the main title)

Author Response

Reviewer 4

Comments and Suggestions for Authors: It is a good revision. I have one comment - please name it a descriptive review (after the main title)

In accordance to reviewer’s comment, we added “a descriptive review” after main title.